# Diurnal variation of motor activity in adult ADHD patients analyzed with methods from graph theory

Ole Bernt Fasmer[1,2,3]*, Erlend Eindride Fasmer[4], Kristin Mjeldheim[5], Wenche Førland[6], Vigdis Elin Giæver Syrstad[1,2,3], Petter Jakobsen[1,3], Jan Øystein Berle[1,2], Tone E. G. Henriksen[2,7], Zahra Sepasdar[8], Erik R. Hauge[1], Ketil J. Oedegaard[1,2,3]

1 Division of Psychiatry, Haukeland University Hospital, Bergen, Norway, 2 Department of Clinical Medicine, University of Bergen, Bergen, Norway, 3 NORMENT, Division of Psychiatry, Haukeland University Hospital, Bergen, Norway, 4 Independent Researcher, Harstad, Norway, 5 Independent Researcher, Hafrsfjord, Norway, 6 Independent Researcher, Stavanger, Norway, 7 Division of Mental Health Care, Valen Hospital, Fonna Local Health Authority, Valen, Norway, 8 School of Electrical and Computer Engineering, Shiraz University, Shiraz, Iran

* ole.fasmer@uib.no

**Data Availability Statement:** All relevant data are within the manuscript and its Supporting Information files.

**Funding:** The authors have received financial support for the research related to this article from

## Abstract

Attention-deficit /hyperactivity disorder (ADHD) is a common neurodevelopmental syndrome characterized by age-inappropriate levels of motor activity, impulsivity and attention. The aim of the present study was to study diurnal variation of motor activity in adult ADHD patients, compared to healthy controls and clinical controls with mood and anxiety disorders. Wrist-worn actigraphs were used to record motor activity in a sample of 81 patients and 30 healthy controls. Time series from registrations in the morning and evening were analyzed using measures of variability, complexity and a newly developed method, the similarity algorithm, based on transforming time series into graphs. In healthy controls the evening registrations showed higher variability and lower complexity compared to morning registrations, however this was evident only in the female controls. In the two patient groups the same measures were not significantly different, with one exception, the graph measure bridges. This was the measure that most clearly separated morning and evening registrations and was significantly different both in healthy controls and in patients with a diagnosis of ADHD. These findings suggest that actigraph registrations, combined with mathematical methods based on graph theory, may be used to elucidate the mechanisms responsible for the diurnal regulation of motor activity.

## Introduction

Attention-deficit /hyperactivity disorder (ADHD) is a common neurodevelopmental psychiatric syndrome characterized by age-inappropriate levels of motor activity, impulsivity and attention [1]. Other symptoms, like sleep problems and drug abuse are also prevalent, and co-morbidity with mood and anxiety disorders is the rule rather than the exception in patients with this disorder [1–3].

A high proportion of adolescents with ADHD maintain the syndrome symptoms in adulthood [1]. In addition, a significant proportion of ADHD patients are first diagnosed as adults,

The Western Norway Regional Health Authority and The Norwegian Resource Center for ADHD, Tourette Syndrome and Narcolepsy. The funders had no role in study design, data collection and analysis, decision to publish, or preparation of the manuscript.

**Competing interests:** The authors have declared that no competing interests exist.

indicating that the developmental trajectory of this syndrome is variable [4]. ADHD in adulthood is associated with adverse occupational outcomes, impaired somatic and psychiatric health, as well as high costs for social and health services [4, 5].

Dysregulated activation is central phenomenon in ADHD [6] and for patients with mood disorders [7]. Objective measures of activity, such as actigraphy may be used to gain more knowledge about the dynamics of activation patterns. The core symptom in ADHD, hyperactivity is diagnostically based on behavioural observations [8], but increased activity levels have been objectively documented by use of actigraphs [9, 10] and infrared motion analyses [11]. Hyperactivity subsides with age [1, 8], but still a study in adults found increased activity to be a more discriminative feature than either inattention or impulsivity [11]. However, objective registrations of activity levels in adult patients have given mixed results [9, 11–13], and there is limited information regarding a more detailed analysis of motor activity patterns in ADHD.

Teicher reported that actigraphically recorded hyperactivity was caused by the absence of quiet periods rather than to periods with overtly increased activity [9]. In a previous report from the same adult clinical sample as in the present paper we found that ADHD patients did not show evidence of hyperactivity, but rather had levels of mean activity similar to normal controls. However, using Fourier analysis with 300 min recordings the ADHD patients had higher power in the high frequency range, corresponding to the period from 2–8 min, and also lower lag1 autocorrelation [14].

In the present paper we further examined the motor activity of adult ADHD patients, using wrist-worn actigraphs, but now with a focus on diurnal variation. Actigraphy has previously mostly been used to assess sleep but is now increasingly being used to record daytime activity and the 24-h rest-activity cycle in patients with psychiatric disorders [15, 16]. In normal controls actigraphically recorded motor activity display a clear diurnal variation (morning vs. evening), both for activity level and variability [16]. ADHD is characterized by increased intraindividual variability both in patients and in animal models of ADHD [17, 18], but in several test systems the diurnal variation of ADHD patients seems to be less than in controls [19, 20].

Biological systems can seldom be described using simple linear models. We have therefore previously employed methods obtained from the fields of non-linear systems and complexity theory to analyze actigraph recordings [14, 21, 22]. In a recent paper we also used a method based on graph theory to analyze time series with motor activity. With this method, called the similarity graph algorithm, time series are mapped into graphs, and the properties of the graphs are analyzed, using familiar techniques from graph theory. In this way more features of the activity can be analyzed. Several methods have been developed in recent years to analyze time series using similar techniques, such as the visibility graph [23], and the horizontal visibility graph methods [24], and these have been applied to diverse fields, ranging from seismology [25] to the study of human walking rhythm [26].

The aim of the present study was to study diurnal variation of motor activity in adult ADHD patients, by comparing actigraphically recorded time series from morning and evening registrations using the similarity graph method as well as traditional mathematical models. Two comparison groups are used, patients with mood and anxiety disorders without a diagnosis of ADHD and normal controls. The hypothesis was that patients with ADHD have a more static activity pattern from morning to evening than controls.

## Materials and methods

### Ethics statement

The Norwegian Regional Medical Research Ethics Committee West approved the study protocol. Written informed consent was obtained from all participants involved in the study.

## Subjects

Patients were recruited from the private psychiatric practice of KM and WF, both certified psychiatrists with long clinical experience. The patients were consecutive new referrals, in need of diagnostic evaluation of either ADHD, or mood/anxiety disorders, and age between 17 and 65 years. Exclusion criteria were inability to speak Norwegian and not being able to comply with the study protocol. A total of 104 patients were recruited. For different reasons (logistics problems, patients forgetting to wear the actigraphs, technical failures) we were not able to obtain complete recordings for all the patients. We were able to analyze a total of 81 recordings, and these are reported on in the present paper. The group consisted of 38 women and 43 men, and the average age was 37.7 ± 11.0 years (mean ± SD), range 17–61. Most of the patients (n = 55) used no psychotropic drugs, of the rest 15 used antidepressants, two lithium, and three other mood stabilizers (lamotrigine or valproate). One patient used a small dose of an antipsychotic drug (olanzapine 2.5 mg), and three used hypnotics or benzodiazepines. Patients using drugs at referral continued unchanged with these during the actigraph recordings.

The control group consisted of 20 women and 10 men, average age 38.7 ± 13.1 years, range 21–66, medical students (n = 5), patients without serious medical or psychiatric symptoms from a primary care office (n = 4) and employees from Bergen University and a psychiatric nursing home (n = 21). None of the control subjects had a history of mood or psychotic disorders. The controls were recruited during a separate study, using the same actigraph equipment as the patients in this study, and are reported on in two previous papers [21, 27].

## Psychiatric assessment

For all diagnostic assessments of the patients a standard clinical interview was used, supplemented when possible with information from collateral sources. The interviews were performed by either KM or WF. The following assessment instruments were also used:

*The Mini-International Neuropsychiatric Interview* (MINIPlus, version 5.0.0), a module based semi-structured interview for DSM-IV and ICD-10 diagnoses [28, 29].

*Montgomery-Asberg Depression Rating Scale* (MADRS), a standard instrument for the assessment of depression [30].

*The Adult ADHD Self-Report Scale* (ASRS). This is the World Health Organization's rating scale designed to measure current symptoms of ADHD in adults. It consists of 18 questions following the DSM-IV criteria for ADHD, with a 5 point scale of 0–4 (0 = never, 4 = very often), and thus a possible range of scores from 0–72. Items 1–9 cover symptoms of inattention and 10–18 hyperactivity and impulsivity [31, 32].

*Wender Utah Rating Scale*, the 25 questions version (WURS-25), a self-rating scale designed to assess symptoms and signs of ADHD in childhood, using a Likert scale of 0–4 (0 = never, 4 = very often), yielding a possible range of scores 0–100 [33]. WURS-25 has been used in previous studies in Norway [31].

*Mood Disorder Questionnaire* (MDQ). This is a screening instrument for bipolar disorder. It is a self-report form that consists of 13 questions scored "Yes" of "No". Positive answer to at least seven questions and confirmation that the symptoms have occurred together and caused problems is suggestive of a bipolar disorder [31, 34].

*Cyclothymic temperament scale* is a self-report form consisting of 21 questions covering the cyclothymic temperament according to the definition of Akiskal. This scale is part of the larger TEMPS-A auto-questionnaire [35–37].

*Hospital Anxiety and Depression Scale* (HADS). This is a self-assessment form for evaluating current depression and anxiety, and has been extensively used, also in Norway [38].

The final diagnostic evaluation, based on DSM-IV criteria, was made after an assessment of all available information, and discussion of each case.

## Recording of motor activity

Motor activity was monitored with an actigraph worn at the right wrist (Actiwatch, Cambridge Neurotechnology Ltd, England). In the actigraph, activity is measured by using a piezoelectric accelerometer that is programmed to record the integration of intensity, amount and duration of movement in all directions. The sampling frequency is 32 Hz and movements over 0.05 g are recorded. A corresponding voltage is produced and is stored as an activity count in the memory unit of the actigraph. The number of counts is proportional to the intensity of the movement. The right wrist was chosen to make the procedure more convenient for the participants, since most of them have their watches around the left wrist and it is cumbersome to have two such devices on the same arm. Previous studies have shown that there are only small differences between the right and left wrist [39, 40].

Total activity counts were recorded for one-minute intervals for a period of six days. The first continuous 24 hour period (0000–2359) for each participant (without missing data due to the actigraph having been removed) was used. From this period we selected morning and evening epochs, each with a 6 hour duration. Morning epochs were defined to occur between 8 AM and 2 PM, and evening epochs between 6 PM and midnight.

An example of actigraph recording is shown in Fig 1.

## Mathematical analyses

For all the time epochs we calculated the mean activity, the standard deviation (SD), expressed as percent of the mean, the root mean square successive differences (RMSSD), also expressed as percent of the mean, and the ratio RMSSD/SD.

In addition we calculated the sample entropy of the time series and a number of measures based on a transformation of the time series into graphs.

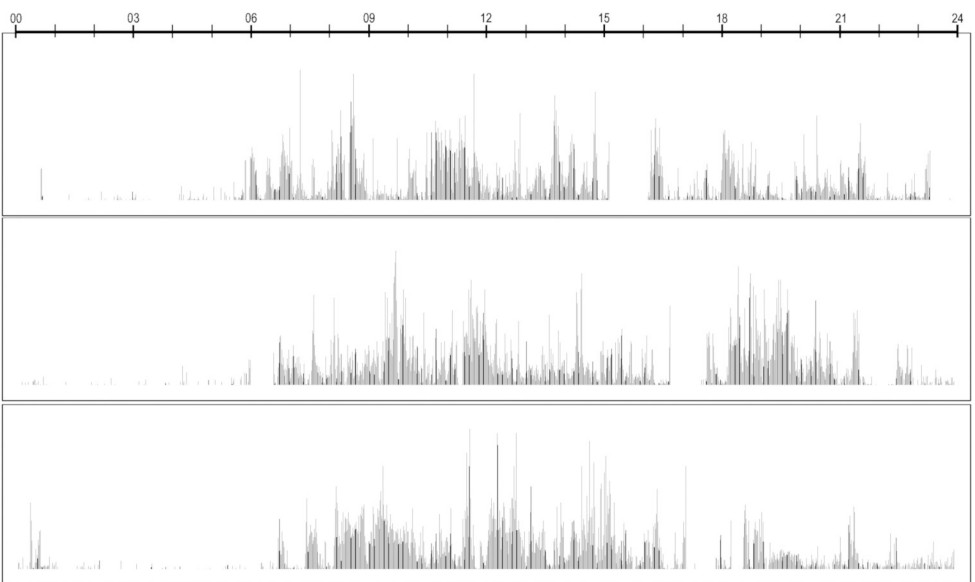

**Fig 1. This is an example of an actigraph registration, motor activity over three 24 hour periods.**

## Sample entropy

Sample entropy is a nonlinear measure, indicating the degree of regularity (complexity) of time series, and is the negative natural logarithm of an estimate of the conditional probability that subseries of a certain length (m = 2) that match point-wise, within a tolerance (r = 0.2), also match at the next point. Sample entropy has been employed for the analysis of different types of biological data since it can be employed with comparatively short time series (>50) and is robust with regard to outliers [41].

## Graph theory

Graph theory is one of the most important and useful branches of mathematics. It is concerned with the study of graphs, which are mathematical structures that model relations between objects.

All definitions in this section can be found in these two references [42, 43]. A *graph* G is an ordered pair (V(G), E(G)) consisting of a set V(G) of nodes and a set E(G), disjoint from V(G), of edges, together with an incidence function $\psi_G$ that associates with each edge of G an unordered pair of (not necessarily distinct) nodes of G. If e is an edge and u and v are nodes such that $\psi_G(e) = \{u, v\}$, then e is said to join u and v, and the nodes u and v are called the ends of e. For simplicity, we write V and E instead of V(G) and E(G).

A graph may exhibit many different topological properties, of which we will only mention the ones relevant to this article. Most of the definitions and concepts in graph theory are suggested by this graphical representation. The ends of an edge are said to be *incident* with the edge, and vice versa. Two nodes which are incident with a common edge are *adjacent*, as are two edges which are incident with a common node, and two distinct adjacent nodes are *neighbours*.

The *degree* of a node v in a graph G, denoted by d(v), is the number of edges of G incident with v. If G is a simple graph, d(v) is the number of neighbours of v in G.

A graph H is called a *subgraph* of a graph G if V(H) $\subseteq$ V(G), E(H) $\subseteq$ E(G), and $\psi_G$ is the restriction of $\psi_G$ E(H).

A *path* is a graph whose nodes can be arranged in a linear sequence in such a way that two nodes are adjacent if they are consecutive in the sequence and are nonadjacent otherwise. Two nodes u and v in a graph G are connected if G contains a u-v path. The graph G itself is *connected* if every two nodes of G are connected.

A connected subgraph H of a graph G is a *connected component* of G if H is not a proper subgraph of a connected subgraph of G.

Let e be an edge of a graph G. If G-e has more connected components than G, then e is a *bridge* of G.

A *complete* graph is a simple graph in which any two nodes are adjacent. A complete subgraph of G is called a *clique* of G. A *k-clique* is a clique with k nodes.

Big O notation is used in computer science to group algorithms according to the growth rate of their running times or space requirements as functions of the input size. We will only consider the running times of the algorithms described in this paper. Although the notation has a precise mathematical definition, the O notation for a function *f* is usually derived with the following two simplification rules. 1. If *f* is a sum of several terms, only the one with the largest growth rate is kept. 2. If *f* is a product of several factors, any constants that do not depend on the input can be removed. To describe the running time of an algorithm with the big O notation is to give an upper bound on its growth rate as a function of the input size. To give some examples, O(1) is the class of algorithms with constant running time, O(log n) is the class of algorithms with logarithmic running time, and O(n) is the class of algorithms with

linear running time. Generally, n represents the number of nodes when it comes to graph algorithms, that is, n = |V|.

## Similarity graph

In this study we apply a heuristic algorithm that is nonlinear and not chaos-based that transforms a time series S = $(x_1, x_2, \ldots, x_n)$ into a *similarity graph* G = (V,E), an undirected graph, where each node $u \in V = \{1, 2, \ldots, n\}$ corresponds to the element $x_u \in S$ and where the node *u* is assigned a weight equal to the value of $x_u$ [44]. The *distance* between two nodes *u* and *v*, is |*u* —*v*|. Two arbitrary nodes *u* and *v* are defined to be *direct neighbours* if their distance is 1. Two nodes *u* and *v* are said to be *similar* to each other (that is, they have a symmetric relationship) if max $(x_u, x_v)$ / min $(x_u, x_v)$ < 1.2. This definition sees the world from the perspective of both *u* and *v*. The undirected similarity graph is constructed as follows. We introduce an undirected edge between two nodes if and only if they are similar to each other and their distance is below a certain threshold *k*. The *k* leftmost and the *k* rightmost nodes are disregarded when counting the number of neighbors of each node such that the node of every considered value in *S* may have as much as 2*k* neighbors. Different values of *k* are giving different similarity graphs.

The choice of 20% as a threshold for defining two points as similar is based on our previous studies of motor activity with the sample entropy method [21, 41]. The sample entropy method is also based on finding points in time series that are similar to one another, and it is customary to use 20% of the standard deviation for defining two points as similar. With the similarity graph method we do not employ the standard deviation, but use 20% of the value of the time points. The standard deviation of these time series from actigraph recordings are usually large, in the order of 100% of the mean, meaning that 20% of the amplitude of an average time point roughly corresponds to 20% of the standard deviation, and we have therefore chosen 20% to define similarity.

We think it is more reasonable to use a criterion that is dependent on actual values of the time series instead of using a fixed value for the whole series (as in sample entropy). One problem with sample entropy is that it is sensitive to outliers, which will increase the standard deviation and thus increase the probability that two points are considered similar, giving a false impression of high regularity. We therefore think it makes more sense to use our definition similarity. The consequence will be that number of points considered similar will be lower, thus avoiding ceiling effects, and will probably increase the ability of the test to detect differences between groups. This will be the case for all the graph measures we use.

The rationale for using a threshold *k* as opposed to always considering connecting a node to every other node of the graph is to compare a value in the time series to its nearest past and nearest future in order to obtain a number (i.e. the number of neighbors) designating how much a given value changes compared to its nearest past and future. The higher degree a certain node has, the more similar its weight is to its *k* preceding and *k* subsequent nodes. The lower degree the node has, the more different its value is from its preceding and subsequent values in the corresponding time series. A node with few or no adjacencies indicates a jump in the activity level, either from low to high activity or vice versa. Repeating the algorithm for different values of *k*, gives different similarity graphs, which may reveal different properties of the underlying time series. Another kind of activity jump is revealed by the graph forming connected components and this implies that two periods of time each have smooth changes in activity internally but that the activity changes in one of them are significantly different from the activity changes in the other one. Yet another kind of activity jump is revealed by the graph forming bridges and these are structures that point in the same direction as connected components but they are not as strong. A graph with one bridge is "almost" two connected

components. The removal of the bridge would result in the graph forming two connected components. Thus bridges reveal activity jumps that are significant but still not as significant as those formed by connected components. Finally another kind of activity jump is revealed by a pair of direct neighbors in an undirected graph missing a relationship meaning that the activity level in one time interval is significantly smaller or larger than the other. The number of $k$-cliques reveal how smooth the activity changes are. The more $k$-cliques the graph exhibits, the more smooth the activity changes are. We have focused on 3-cliques in this article. Every 3-clique reveals that a tuple of three nodes have similar values. The running time of the algorithm is O(|V|) [45]. We have computed the connected components of the graph with a depth-first search in O(|E|) time [45]. The 3-cliques are computed with a O(a(G)m) time algorithm designed by Chiba and Nishizeki, where a(G) is the arboricity of the graph [46].

In our previous paper on graph analyses of motor activity of patients with schizophrenic and depression [44], using 300 min sequences, values of k from 20 up to 80 seemed to give comparable results. In preliminary analyses of the present dataset we found that 40 + 40 neighbours gave the best separation between morning and evening sequences for the control group using the different graph measures. We have therefore presented the results for analyses based on 40 + 40 neighbours, and we have calculated the following measures: The number of edges, components, bridges, cliques (3-cliques), the maximum number of edges, the number of nodes with zero edges (i.e. connected components consisting of a single node) and the number of missing edges between nearest neighbors. The values for cliques are transformed to ln cliques. For number of bridges we have also calculated the values using different number of edges, from 2 + 2 and up to 80 + 80.

An example of a similarity graph is shown in Fig 2.

## Statistics

Independent samples or paired samples t-test were used to compare two groups with continuous data and Chi square test for categorical data. One-way ANOVA was used to evaluate differences between three groups, with post hoc Bonferroni tests. The effect of age on cyclothymic temperament (CT)-scores was evaluated using Pearson correlations, and Pearson correlations were also calculated for the relation between bridges, SD and sample entropy. The effects of age and gender on the differences between groups and on differences between morning and evening registrations were evaluated using analysis of covariance (ANCOVA). Effect sizes (Cohens d) are indicated where relevant. Linear mixed models (LMM) were employed to test for interaction between time (morning vs. evening) and diagnosis (ADHD or not ADHD vs. healthy controls). SPSS version 25 was used for the statistical analyses.

## Results

Table 1 shows the clinical characteristics of the clinical sample according to the presence or not of ADHD. Fortytwo patients received a diagnosis of ADHD while 39 patients did not have ADHD. Age and gender distribution were not significantly different between the groups, the only differences were in higher WURS- and ASRS-scores in the ADHD patients.

Table 2 shows the results of actigraph recordings in the morning. With one-way ANOVA three measures showed differences between the three groups, SD (p = 0.046), RMSSD (p = 0.015) and missing edges (p = 0.008). After Bonferroni corrections two measures were different between ADHD and healthy controls, RMMSD and number of missing edges, and for both measures the values were higher in the ADHD group. For the evening registrations (Table 3) two measures were different between groups, the RMSSD/SD ratio (p = 0.027) and bridges (p = 0.047). After Bonferroni corrections only RMSSD/SD was different between

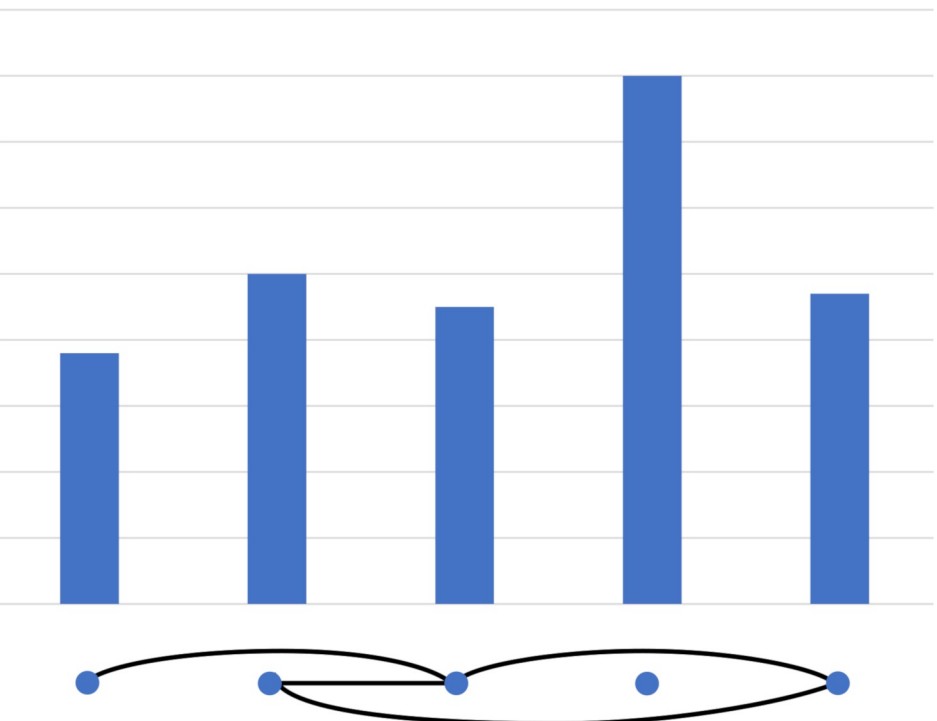

**Fig 2. This is an example of a $k$ = 2 time series (five poles) converted to a graph (five dots, below, with edges as solid lines).** This also illustrates the concepts of components (two components; the first consist of 1,2,3 and 5, the second of 4), bridges (one bridge, the edge between 1 and 3) and cliques (one 3-clique: 2,3 and 5).

ADHD and healthy controls (higher value in the ADHD group). Table 4 shows differences between morning and evening registrations, separately for healthy controls and ADHD patients. For healthy controls 9 of 12 measures differed between morning and evening registrations, while for ADHD patients only one measure, bridges, was different. The strongest difference, measured with effect size, Cohens d, between morning and evening registrations was for

**Table 1. Characteristics of the clinical sample according to the presence or not of ADHD.**

|  | ADHD (n = 42)* | Not ADHD (n = 39)* | P |
|---|---|---|---|
| Age (mean ± SD) | 37.4 ± 10.6 | 38.0 ± 11.5 | 0.801 |
| Gender (m/f) | 24/18 | 19/20 | 0.448 |
| WURS | 51.1 ± 19.7 | 29.4 ± 15.9 | <0.001 |
| ASRS | 47.6 ± 13.1 | 33.5 ± 12.4 | <0.001 |
| HADS |  |  |  |
| Depression | 4.5 ± 3.8 | 5.0 ± 4.1 | 0.527 |
| Anxiety | 9.5 ± 4.6 | 9.19 ± 4.7 | 0.724 |
| MADRS | 13.5 ± 7.6 | 13.7 ± 8.3 | 0.928 |
| MDQ | 6.8 ± 3.9 | 6.4 ± 3.6 | 0.396 |
| CT | 11.6 ± 4.7 | 11.6 ± 3.9 | 0.974 |
| CT ≥11 | 57% (18/42) | 61% (22/36) | 0.722 |

Chi-square test for gender and CT ≥11, for the other measures independent samples t-test.

* Number of subjects varies somewhat between the different measures, n = 40 to 42 for ADHD and n = 35 to 39 for not ADHD.

**Table 2. Actigraphic registrations in the morning, 360 min (08–14).** Controls and the clinical group divided according to the presence or not of ADHD. For the graph analyses the number of neighbours is 40 + 40.

|  | Controls | ADHD | Not ADHD | ANOVA |
|---|---|---|---|---|
|  | (n = 30) | (n = 42) | (n = 39) |  |
| Mean | 385 ± 178 | 289 ± 152 | 312 ± 192 | F (108,2) = 2.774, P = 0.067 |
| SD (% of mean) | 105 ± 31 | 131 ± 49 | 126 ± 50 | F (108,2) = 3.167, P = 0.046 |
| RMSSD (% of mean) | 86 ± 17 | 110 ± 37* | 107 ± 42 | F (108,2) = 4.484, P = 0.013 |
| RMSSD/SD | 0.85 ± 0.12 | 0.86 ± 0.16 | 0.86 ± 0.13 | F (108,2) = 0.132, P = 0.876 |
| Edges | 7.94 ± 2.80 | 6.49 ± 2.96 | 7.07 ± 3.52 | F (108,2) = 1.879, P = 0.158 |
| Components | 104 ± 35 | 135 ± 68 | 132 ± 68 | F (108,2) = 2.613, P = 0.078 |
| Bridges | 38.6 ± 8.5 | 36.1 ± 11.5 | 36.6 ± 8.6 | F (108,2) = 0.610, P = 0.545 |
| Missing edges | 309 ± 14 | 321 ± 16* | 318 ± 17 | F (108,2) = 5.075, P = 0.008 |
| Max number of edges | 21.8 ± 5.1 | 20.8 ± 7.1 | 22.0 ± 7.7 | F (108,2) = 0.359, P = 0.699 |
| Nodes with zero edges | 117 ± 34 | 144 ± 62 | 143 ± 62 | F (108,2) = 2.579, P = 0.081 |
| Ln cliques | 7.69 ± 0.68 | 7.32 ± 0.90 | 7.52 ± 1.03 | F (108,2) = 1.518, P = 0.224 |
| Sample entropy | 0.98 ± 0.46 | 0.76 ± 0.45 | 0.74 ± 0.44 | F (108,2) = 2.945, P = 0.057 |

* P < 0.05, ADHD vs. controls for Bonferroni test.

bridges in the healthy controls (Cohens d 1.34). In ADHD patients this value was 0.83. Using linear mixed model analyses to test for interactions between diagnosis (healthy controls vs. ADHD) and time (morning vs. evening) three measures were significant, mean activity level (p = 0.039), SD (p = 0.039) and sample entropy (p = 0.014).

S1 Table is similar to S4 Table, but in this table healthy controls are compared to patients without a diagnosis of ADHD. For the not ADHD clinical group none of the morning/evening differences were significant, and with the linear mixed model interactions analyses between diagnosis (healthy controls vs. not ADHD) and time (morning vs. evening) two measures were significant, bridges (p = 0.018) and sample entropy (p = 0.014).

**Table 3. Actigraphic registrations in the evening, 360 min (18–24).** Controls and the clinical group divided according to the presence or not of ADHD. For the graph analyses the number of neighbours is 40 + 40.

|  | Controls | ADHD | Not ADHD | ANOVA |
|---|---|---|---|---|
|  | (n = 30) | (n = 41) | (n = 39) |  |
| Mean | 309 ±139 | 299 ± 172 | 257 ± 139 | F (106,2) = 1.188, P = 0.309 |
| SD (% of mean) | 137 ± 41 | 134 ± 41 | 139 ± 51 | F (106,2) = 0.140, P = 0.870 |
| RMSSD (% of mean) | 104 ± 31 | 111 ± 35 | 116 ± 51 | F (106,2) = 0.729, P = 0.485 |
| RMSSD/SD | 0.76 ± 0.08 | 0.84 ± 0.14* | 0.84 ± 0.15 | F (106,2) = 3.743, P = 0.027 |
| Edges | 6.27 ± 2.54 | 6.25 ± 2.75 | 5.87 ± 2.74 | F (106,2) = 0.248, P = 0.781 |
| Components | 146 ± 58 | 140 ± 54 | 151 ± 63 | F (106,2) = 0.354, P = 0.703 |
| Bridges | 25.1 ± 11.5 | 26.8 ± 10.9 | 31.7 ± 12.1 | F (106,2) = 3.158, P = 0.047 |
| Missing edges | 317 ± 19 | 321 ± 14 | 322 ± 15 | F (106,2) = 0.799, P = 0.453 |
| Max number of edges | 21.8 ± 7.7 | 20.1 ± 5.5 | 20.5 ± 5.8 | F (106,2) = 0.626, P = 0.536 |
| Nodes with zero edges | 142 ± 50 | 141 ± 46 | 154 ± 57 | F (106,2) = 0.740, P = 0.480 |
| Ln cliques | 7.33 ± 0.86 | 7.25 ± 0.79 | 7.23 ± 0.84 | F (106,2) = 0.158, P = 0.854 |
| Sample entropy | 0.52 ± 0.28 | 0.66 ± 0.41 | 0.62 ± 0.39 | F (106,2) = 1.328, P = 0.269 |

* P< 0.05, ADHD vs. controls for Bonferroni test.

**Table 4. Actigraphic registrations in the morning and evening, 360 min (08–14 and 18–24).** Healthy controls (n = 30) and ADHD patients (n = 41).

| | Healthy controls | | | | ADHD | | | | | | |
|---|---|---|---|---|---|---|---|---|---|---|---|
| | Morning | Evening | P | d# | Morning | Evening | P | d | LMM* | | |
| | | | | | | | | | D | T | D x T |
| Mean | 385±178 | 309±139 | 0.049 | 0.48 | 289±152 | 299±172 | 0.536 | 0.06 | 0.012 | 0.204 | 0.039 |
| SD | 105±31 | 137±41 | 0.003 | 0.88 | 131±49 | 134±41 | 0.868 | 0.07 | 0.015 | 0.422 | 0.039 |
| RMSSD | 86±17 | 104±31 | 0.014 | 0.72 | 110±37 | 111±35 | 0.991 | 0.03 | 0.014 | 0.488 | 0.107 |
| RMSSD/SD | 0.85±0.12 | 0.76±0.08 | 0.003 | 0.88 | 0.86±0.16 | 0.84±0.14 | 0.548 | 0.13 | 0.514 | 0.821 | 0.155 |
| Edges | 7.94±2.80 | 6.27±2.54 | 0.022 | 0.63 | 6.46±2.99 | 6.25±2.75 | 0.711 | 0.08 | 0.075 | 0.684 | 0.138 |
| Components | 104±35 | 146±58 | 0.001 | 0.88 | 135±68 | 140±54 | 0.765 | 0.08 | 0.028 | 0.517 | 0.057 |
| Bridges | 38.6±8.5 | 25.1±11.5 | <0.001 | 1.34 | 36.1±11.5 | 26.8±10.9 | <0.001 | 0.83 | 0.217 | 0.050 | 0.231 |
| Missing edges | 309±14 | 317±19 | 0.062 | 0.48 | 321±16 | 321±14 | 0.932 | 0.00 | 0.014 | 0.433 | 0.111 |
| Max edges | 21.8±5.1 | 21.8±7.7 | 0.984 | 0.00 | 20.8±7.1 | 20.1±5.5 | 0.623 | 0.11 | 0.907 | 0.650 | 0.761 |
| Zero edges | 117±34 | 142±50 | 0.030 | 0.58 | 144±62 | 141±46 | 0.744 | 0.05 | 0.045 | 0.328 | 0.096 |
| Ln cliques | 7.69±0.68 | 7.33±0.86 | 0.093 | 0.46 | 7.32±0.90 | 7.25±0.79 | 0.717 | 0.08 | 0.124 | 0.793 | 0.287 |
| Sample entropy | 0.98±0.46 | 0.52±0.28 | <0.001 | 1.21 | 0.76±0.45 | 0.66±0.41 | 0.292 | 0.23 | 0.019 | 0.399 | 0.014 |

M: linear mixed model, diagnosis = ADHD vs. healthy controls, time = morning vs. evening, D = diagnosis, T = time, D x T = interaction diagnosis and time.
#d: effect size (Cohen).

In the ADHD group, 21 patients had CT and 18 had not CT. S2 Table shows differences between morning and evening registrations for ADHD patients with and without cyclothymic temperament, showing that only bridges were different (p = 0.012 for both CT and not CT).

S3 Table shows calculations of bridges using 6 different values for neighbours, and comparing morning with evening for healthy controls and ADHD patients separately. For all these values of neighbours the difference between morning and evening was higher for the healthy controls than for the ADHD patients. The effect size was highest for 40 + 40 (1.34, healthy controls) and 80 + 80 (1.64, healthy controls). For bridges (40 + 40 neighbours) there was a modest negative correlation with SD (-0.331, p<0.001) and a positive correlation with sample entropy (0.382, p < 0.001).

There was a small negative correlation between age and score on the CT scale (-0.288 p = 0.013), and an interaction with borderline significance (F = 3.898, p = 0.050) between age and time of day (morning vs. evening) for the sample entropy measure in the healthy controls. Apart from this there were no significant effects of age.

In the morning registrations there was an interaction between gender and diagnosis for the sample entropy measure. For the controls sample entropy is higher in females than in males, while for ADHD and not ADHD there were no significant differences between the genders (S4 Table).

In the evening registrations there were significant effects of gender for 8 of the 12 measures; SD, RMSSD, edges, components, max edges, zero edges, ln cliques, and sample entropy. These results are shown in S5 Table. Values for all the 12 measures separately for males and females are shown in S6 and S7 Tables.

In the comparison between morning and evening registrations there were no significant effects of gender in the healthy controls, as shown in S8 Table. Values for all the 12 measures separately for males and females are shown in S9 Table. For males only one measure, bridges, was significantly different between morning and evening registrations, while for females 8 measures were significantly different.

In the comparison between morning and evening registrations there were significant effects of gender for 4 of the 12 measures in the ADHD group, SD, edges, components and sample entropy, as shown in S10 Table. Values for all the 12 measures separately for males and females are shown in S11 Table. For males only one measure, bridges, was significantly different between morning and evening registrations, while for females none of the measures were significantly different.

In the comparison between morning and evening registrations for the not ADHD group there were no effects of gender.

## Discussion

We found a clear morning to evening difference in the healthy controls for most actigraphic measures. Contrasting this, both the ADHD and the non-ADHD patient groups, all but one measure did not show significant morning to evening variation. The exception was the graph theoretical concept bridges. The second main finding was that bridges was the measure that most clearly separates morning and evening registrations. This diurnal difference was significant both for healthy controls as well as for the ADHD patients.

In the morning registrations two measures, RMSSD and missing edges, were significantly different in the ADHD group compared to healthy controls. RMSSD was higher in ADHD patients, and there was also a higher number of missing edges in the ADHD patients compared to healthy controls, meaning that more nodes (points in the time series) are different from the nodes immediately before and after the index node. Missing edges is therefore a graph theoretical measure that gives information similar to that of RMSSD.

The SD was also higher in the ADHD group than in healthy controls, but the difference between the three groups was not significant using ANOVA. These findings are consistent with a number of previous studies showing that higher intraindividual variability is a characteristic feature of ADHD [17, 47–49], and is also found in animal models of ADHD [18]. However, variability, measured with the SD, is an important finding in a wide range of other conditions and disorders as well [50, 51].

In the evening registrations only one measure, the ratio RMSSD/SD, was different in ADHD patients compared to healthy controls, while there were only small differences in RMSSD, SD and missing edges. Interestingly, this shares similarities with the activity profiled of manic patients, as shown in a study by Krane-Gartiser et al. [22], reporting higher RMSSD/SD in manic patients compared to controls, in 64 min recordings both in the morning and evening. Increased RMSSD/SD ratio has also been found in actigraphic studies of schizophrenic patients [21].

However, there were only small differences between the ADHD group and the clinical controls, mostly patients with mood and anxiety disorders, indicating that the differences in variability are not specific to a diagnosis of ADHD.

When comparing morning and evening recordings in healthy controls there was a striking difference for most of the measures, both with regard to activity level and variability. In general the variability measures were higher in the evening. Number of edges was also lower, meaning more differences between an index node and the neighbours, consistent with a higher variability. The number of components was higher in the evening, indicating that there was more often a change in activity to a lower or a higher level, again consistent with a general higher variability in the evening. Apparently in contrast to this, a measure of complexity, sample entropy, was significantly lower in the evening. With regard to way sample entropy is calculated there is no obvious reason why we have found such an inverse relation between this measure and ordinary variability measures. However, the association between high variability,

measured with RMSSD and SD, and low complexity measured with sample entropy, has also been found in another study of motor activity [21]. Reduced complexity has been associated with disease states and aging [52], and in a study of schizophrenic patients motor activity complexity was found to be lower in evening than in the morning [16]. But the present study shows that changes in levels of complexity also may be a feature of normal diurnal variation.

In contrast, both in ADHD patients and in clinical controls (not ADHD), there is little difference between morning and evening recordings, the one exception being bridges. For ADHD patients the difference in number of bridges is substantial, with lower values in the evening, while for the not ADHD group it is smaller, but higher than for the other measures.

The number of bridges is even lower in the evening in the healthy controls, and this measure is the most highly discriminant of all (d = 1.34). There is a lower number of bridges in evening for all choices of neighbours, from 2 +2 and up to 80 + 80, although the largest difference is for 40 + 40 and 80 + 80 neighbours. However, it is not easy to compare this measure to the others we have used, but even though we do not have a clear explanation for these findings concerning bridges, we think that such results may give clues to the underlying dynamics of the time series, and thus the regulation of motor activity.

In a previous study of the same sample [17] we found differences in scores on a neuropsychological test (CPT) when comparing the ADHD patients with and without a cyclothymic temperament. Mood instability, as reflected in the scores on the CT scale, is an important part of the clinical picture of ADHD, but concerning variation between motor activity patterns in the morning and evening there seem to be no differences between patients with and without CT.

In the present study we have found that in ADHD patient motor activity is more stable across the day compared to controls. This is in accordance with the findings of Boonstra et al. [19] that the rhythm measure interdaily stability is higher, and intradaily variability is lower in ADHD patients than in controls indicating a more repeatable daily activity schedule and more consolidated rest and activity periods. Additionally, there are other studies that have found reduced diurnal variation in other functions as well, such as the study of Hunt et al. [53] on cognitive function in young adults with ADHD. The participants lacked the normal significant diurnal variation in neuropsychological performance. Baird et al. [20] looked at actigraphy data, clock genes, salivary melatonin and cortisol secretion in adult ADHD-patients compared to controls. For actigraphy and cortisol secretion they found a significant rhythmicity both in ADHD and controls. However, clock gene expression (PER2) in the oral mucosa exhibited a significant rhythm in controls, which was lost in ADHD patients. Similarly for salivary melatonin, there was a significant rhythm in controls, but a loss of this rhythmicity in the ADHD group. Reduced diurnal variation is also found in actigraphic studies of manic patients, both with regard to activity level, SD and RMSSD [16]. On the other hand, in a study of heart rate variability [54] in children with ADHD as well as normal controls showed pronounced variation from morning to evening in measures related to vagus activity.

Both motor activity, arousal and other biological functions are regulated by endogenous chronobiological rhythms, the most important being the circadian [55]. Loss of diurnal fluctuations in physiological outputs could be an expression of weak circadian rhythms, or in other words, reduced amplitude of circadian function. Reduced circadian amplitude has been found in schizophrenia and affective disorders, and is associated with reduced cognitive function, affective symptoms and subjectively reported wellbeing [56, 57]. There is a high comorbidity and many shared clinical traits between bipolar disorder and ADHD, and also involvement of similar neurotransmitter systems, including dopamine. The stable activity patterns from morning to evening for the patients with ADHD may as well reflect the presence of two consecutive endogenous days with similar activity structures, and is compatible with the fact that in

ADHD substantial evidence point to changes in dopaminergic systems [58]. Another reason for loss of diurnal variation could be the possibility of bifurcation of biological rhythms in susceptible individuals [59]. Bifurcation means switching the circadian rhythm from a 24-h cycle to 12-h SCN-generated cycles, and can be provoked in hamsters and mice in extreme light cycles [60]. Human data from bipolar patients in a manic episode support this hypothesis [61].

The recent advances in theories and research on biological rhythms is promising and could lead to better understanding of the mechanisms involved. There were only small effects of age in the present sample of patients and controls, but for gender there were in part large and potential interesting differences between males and females. These differences were found primarily in the evening registrations, but are most conspicuous when looking at the differences between morning and evening for the healthy controls. There were significant differences between morning and evening for females, but not for males, again with the interesting exception of bridges. In females variability in general was higher in the evening, while total level of motor activity is unchanged and complexity was lower (lower sample entropy values). For the ADHD patients there were no significant morning vs. evening differences for either gender, with the exception of bridges for males. We are not aware of similar studies on gender differences in actigraphic measures, and it is therefore difficult to draw conclusions, but regarding the increased variability in motor activity that is previously found in depressed patients [62] one interesting hypothesis may be that the increased variability in the evening recordings in females may be related to the known increased propensity to depression in females [63].

## Limitations

It is always possible that treatment with psychotropic medication may have influenced results in studies such as this, but the majority of our patients did not use psychotropic medication.

We have not looked at sleep parameters, and sleep problems is of course common both in patients with ADHD and mood disorders [64, 65]. This may have influenced results, but it is difficult to separate such effects from other effects on rest and activity rhythms. The patients in this study were all outpatients, but we have not access to information that may allow us to compare their activity schedules to those of the controls. This type of bias is therefore difficult to evaluate. Diagnoses were assessed non-blind, but the actigraphic registrations and mathematical analyses did not require subjective evaluations. Participants were asked to remove the actigraphs while showering or taking a bath, but all the registrations were inspected manually to avoid such periods when extracting morning and evening sequences.

Concerning the effects of gender, it is important to note that the groups are rather small, particularly for the healthy controls, making the results more uncertain than for the groups containing both genders.

## Conclusions

In the present study we have used actigraph registrations combined with mathematical methods, including the similarity graph algorithm, to study diurnal variation of motor activity. In healthy controls most of the actigraphic measures showed clear differences between morning and evening registrations, in particular higher variability and lower complexity in evening registrations compared to the morning. However this was evident only in the female controls. In a mixed sample of outpatients the same measures were not significantly different, with one exception, the graph theoretical measure bridges. This was the measure that most clearly separated morning and evening registrations and this diurnal difference was significant both in healthy controls and in patients with a diagnosis of ADHD. We suggest that by combining different mathematical methods, including the present measures based on graph theory, analyses

of actigraph registrations may give clues to the underlying dynamics of the time series, and thus shed light on the regulation of motor activity in ADHD as well as in other conditions.

## Supporting information

**S1 Table. Actigraphic registrations in the morning and evening, 360 min (08–14 and 18–24).** Healthy controls and clinical controls (Not ADHD).
(DOCX)

**S2 Table. Actigraphic registrations in the morning and evening, 360 min (08–14 and 18–24).** ADHD patients with (n = 24) and without cyclothymic temperament (n = 17).
(DOCX)

**S3 Table. Actigraphic registrations in the morning and evening, 360 min (08–14 and 18–24).** ADHD patients (n = 42/41*) and healthy controls (n = 30). Number of bridges for different choices of neighbours.
(DOCX)

**S4 Table. Effect of gender on actigraphic registrations in the morning, 360 min (18–24) using analysis of covariance (ANCOVA).**
(DOCX)

**S5 Table. Effect of gender on actigraphic registrations in the evening, 360 min (18–24) using analysis of covariance (ANCOVA).**
(DOCX)

**S6 Table. Actigraphic registrations in the evening, 360 min (18–24) for males.** Controls and the clinical group divided according to the presence or not of ADHD. For the graph analyses the number of neighbours is 40 + 40.
(DOCX)

**S7 Table. Actigraphic registrations in the evening, 360 min (18–24) for females.** Controls and the clinical group divided according to the presence or not of ADHD. For the graph analyses the number of neighbours is 40 + 40.
(DOCX)

**S8 Table. Effect of gender on actigraphic registrations in the morning and evening, 360 min (18–24) using analysis of covariance (ANCOVA).**
(DOCX)

**S9 Table. Actigraphic registrations in the morning and evening, 360 min (08–14 and 18–24).** Healthy controls, results for males and females separately.
(DOCX)

**S10 Table. Effect of gender on actigraphic registrations in the morning and evening, 360 min (18–24) using analysis of covariance (ANCOVA).**
(DOCX)

**S11 Table. Actigraphic registrations in the morning and evening, 360 min (08–14 and 18–24).** ADHD patients.
(DOCX)

**S1 File. Background data.**
(XLSX)

## Author Contributions

**Conceptualization:** Ole Bernt Fasmer, Erlend Eindride Fasmer, Kristin Mjeldheim, Wenche Førland, Jan Øystein Berle, Ketil J. Oedegaard.

**Data curation:** Ole Bernt Fasmer, Kristin Mjeldheim, Wenche Førland, Jan Øystein Berle, Erik R. Hauge.

**Formal analysis:** Ole Bernt Fasmer, Erlend Eindride Fasmer.

**Investigation:** Ole Bernt Fasmer, Kristin Mjeldheim, Wenche Førland, Jan Øystein Berle.

**Methodology:** Erlend Eindride Fasmer, Zahra Sepasdar, Erik R. Hauge.

**Project administration:** Ole Bernt Fasmer.

**Software:** Erlend Eindride Fasmer.

**Writing – original draft:** Ole Bernt Fasmer, Erlend Eindride Fasmer, Kristin Mjeldheim, Wenche Førland, Tone E. G. Henriksen, Zahra Sepasdar.

**Writing – review & editing:** Ole Bernt Fasmer, Erlend Eindride Fasmer, Kristin Mjeldheim, Wenche Førland, Vigdis Elin Giæver Syrstad, Petter Jakobsen, Jan Øystein Berle, Tone E. G. Henriksen, Zahra Sepasdar, Erik R. Hauge, Ketil J. Oedegaard.

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
