## [Decision Letter · Decision Letter 0]

17 Aug 2020

PONE-D-20-19305

Diurnal variation of motor activity in adult ADHD patients analyzed with methods from graph theory

PLOS ONE

Dear Dr. Ole Bernt Fasmer,

Thank you for submitting your manuscript to PLOS ONE. After careful consideration, we feel that it has merit but does not fully meet PLOS ONE’s publication criteria as it currently stands. Therefore, we invite you to submit a revised version of the manuscript that addresses the points raised during the review process.

We look forward to receiving your revised manuscript.

Kind regards,

Pan Lin

Academic Editor

PLOS ONE

Journal Requirements:

2.We note that you have indicated that data from this study are available upon request. PLOS only allows data to be available upon request if there are legal or ethical restrictions on sharing data publicly. For information on unacceptable data access restrictions, please see http://journals.plos.org/plosone/s/data-availability#loc-unacceptable-data-access-restrictions.

Reviewers' comments:

Reviewer's Responses to Questions

**Comments to the Author**

1. Is the manuscript technically sound, and do the data support the conclusions?

Reviewer #1: Yes

Reviewer #2: Partly

2. Has the statistical analysis been performed appropriately and rigorously? 

Reviewer #1: Yes

Reviewer #2: Yes

3. Have the authors made all data underlying the findings in their manuscript fully available?

Reviewer #1: No

Reviewer #2: No

4. Is the manuscript presented in an intelligible fashion and written in standard English?

Reviewer #1: Yes

Reviewer #2: Yes

5. Review Comments to the Author

Reviewer #1: This paper focuses on the diurnal variation of motor activity in normal, ADHD and clinic not ADHD people, as well as their comparisons. Authors used a portable actigraphy to record the motor series and tried to character motor feature of ADHD by adopting 12 measures. They reported that in normal people, the evening activity show higher variability and less complexity than morning activity, but it is not observed in ADHD. They also considered the effects of age, gender, cyclothymic temperament on the results. In my opinion, using a portable actigraphy to study motor function has a promising future in clinical diagnosis, exploring neural basis and so on, and authors also obtained some interesting results. But before the publication, there are many problems need to be addressed.

1. Can authors declare the difference between this paper and Ref.17?

2. It is really hard for me to understand how a graph is built from a single motor series

3. Discussion needs to be majorly revised. Many descriptions are actually results (e.g., lines 452-456, 464-468). And many sentences are meaningless (e.g., lines 448, 461-463). Most importantly, many measures can characterize the diurnal difference, but their meanings for patients are not clear explained. For example, in lines 425-426, the description on RMSSD is actually a definition, not its meaning for ADHD. I suggest authors pay more effort to dig deeper meaning of this paper.

4. I notice that many results are not provided in a table (e.g., lines 393-398) and just written in main text. I suggest that a summarize on results in a table is more clear for readers.

5. In graph theory, many measures are given, but the degree, path, acyclic and spanning subgraph are not used in later analysis. If true, please delete them.

6. CT is cyclothymic temperament? Please identify the abbreviation while using it first time.

7. Line 323, 39 should be 39 patients or something.

8. The grammatical tense is not consistent, e.g., lines338 uses ‘was’, line 342 uses ‘is’, pleases unite them through the paper.

9. Line 344, what does the (1.34) mean.

10. After tables, *p=0.05 should be p<0.05.

11. Line 385, p=0.05 is not significant, pleases revise the description

12. Lines 397-398, I didn’t found the results corresponding to the description.

13. Line 404, I know that 8/12 indicates only 8 measures are significant, but it is really a strange description, can we just use 8?

14. Line 483, authors should present the current results and then compare them to previous.

15. Line 500, is it a replicated description ‘differently put; reduced amplitude….. Reduced….’

16. In conclusion, I suggest authors don’t use newly, because it is hard to identify what is really new.

17. Finally, can author explain to me why we have a higher variability and lower complexity in the evening, does it relate to our brain in some aspects?

Reviewer #2: Based on previous studies, the authors tested the assumption that diurnal variation of motor activity of ADHD patients is less than normal controls. With measures from variability, complexity and graph theory aspects, they conducted analysis of wrist-worn actigraphs time-series from ADHD patients, normal controls and clinical controls with mood and anxiety disorders. Their results suggested most of the measures being different between morning and evening in normal controls are not significantly different in both ADHD patients and non-ADHD clinical controls, but “bridge” is significantly different in ADHD and not different in non-ADHD. These results largely support their assumptions about reduced diurnal variation in ADHD. In addition, the exception of changes in bridge may suggest specificity about the diurnal regulation of motor activity in ADHD patients, when comparing with other mental disorders showing diurnal regulation disorders.

However, I have several concerns regarding to the results that should be clarified by authors.

Major 1: During the construction of similarity graph, the similar was detected by max(ux,uv)/min(ux,uv)<1.2. The authors explained this definition with analogy to the tolerance in sample entropy. However, unlike sample entropy where the tolerance range is a fix value (0.2*std), this definition will lead to a flexible range depending on the amplitude of the denominator. In other word, smaller signal value in the bottom will have narrower tolerance range and harder to “be similar” to others. The authors should explain why they use this criteria and what will be the consequence in the following computation of graph measures.

Minor 1: For readability, I suggest the authors to give a schematic illustration for the construction of similarity graph on time-series. In addition, authors can show example of actigraphs time-series. In addition, different kinds of activity jumps detected by graph measures, degree, bridge, and k-clique detect should be given to guide understanding.

Minor 2: The discussion about diurnal fluctuations and weak circadian rhythms, etc are too far away from the topic and the presented evidences, given the fact that the authors cannot explain what is told by the bridge changes.

Minor 3: Not clear why in healthy controls the evening registrations showed higher variability and lower complexity compared to morning registrations. In principle sample entropy is associated with measures of variability. The authors can recheck.

6. PLOS authors have the option to publish the peer review history of their article (what does this mean?). If published, this will include your full peer review and any attached files.

Reviewer #1: No

Reviewer #2: No

---

## [Author Response · Author response to Decision Letter 0]

25 Sep 2020

Response to Reviewers

Reviewer # 1 :

1. Can authors declare the difference between this paper and Ref.17

Response :

In Reference 17 (Fasmer OB, Mjeldheim K, Førland W, Hansen AL, Syrstad VEG, Oedegaard KJ, et al. Linear and non-linear analyses of Conner’s Continuous Performance Test-II discriminate adult patients with attention deficit hyperactivity disorder from patients with mood and anxiety disorders), we did use the same patient sample as in the present study, and furthermore divided tha sample into patients with and without ADHD. However, in Reference 17 we have looked at time series from the the CPT-test, and not from motor activity. In addition, cyclothymic temperament was a cental focus in that paper. In the present paper, besides using time series of motor activity, we study diurnal variation and used methods from graph theory to analyze the time series, in addtion to mathematical methods also utilized in reference 17.

2.It is really hard for me to understand how a graph is built from a

single motor series

Response :

Development of our similarity graph algorithm is based on work done by Lacasa et al (reference 23, Lacasa L, Luque B, Ballesteros F, Luque J, Nuño JC. From time series to complex networks: The visibility graph), where the theoretical background for transforming time series into graphs is described. 

Motor activity time series is only one example of time series that can be trasformed into graphs, and in such a graph every node corresponds to a point in the time series, and two nodes are connected if the values of the motor activity are similar.

To explain in more detail : The similarity graph algorithm transforms time series into undirected similarity graphs, were two time points within a certain distance k connects with an edge if they correspond to each other within a given tolerance. The k preceding and the k succeeding time points of a random point in the time series defines to be nearest neighbors of such a time point. The threshold for similarity between two time points (u and v) defines as max (xu, xv) / min (xu, xv) < 1.2. Any time point with few or no connections (edges) to its nearest neighbors indicates an alteration in activity

3.Discussion needs to be majorly revised. Many descriptions are actually

results (e.g., lines 452-456, 464-468). And many sentences are meaningless

(e.g., lines 448, 461-463). Most importantly, many measures can

characterize the diurnal difference, but their meanings for patients are

not clear explained. For example, in lines 425-426, the description on

RMSSD is actually a definition, not its meaning for ADHD. I suggest authors

pay more effort to dig deeper meaning of this paper.

Response :

We have altered the parts of the discussion mentioned.

Concerning lines 425-426 : This must be seen in relation to the rest of this paragraph and the next. We agree with the reviewer that these parts need to be revised, and have made alterations from line 425 to line 440.

Concerning line 448: We have deleted the sentence that starts in line 448.

Concerning lines 452-456: We have made several alterations to this part.

Concerning lines 461-463: We have altered this sentence, and moved it some lines up in the same paragraph, hopefully making the meaning clearer.

Concerning lines 464-468:We have deleted the listing of results in the last sentence of this paragraph.

Concerning the deeper meaning of this paper: The changes we have made may hopefully give more meaning to the results we have presented. However, as we have stated in the conclusion section of the discussion, the present findings do not in any way fully explain the underlying dynamics of the diurnal regulation of motor activity, and activation in general. However, we think the present study may indicate the way foreward, namely using actigraph registrations in natural situations combined with mathematical methods to further unravel these mechanisms. 

4.I notice that many results are not provided in a table (e.g., lines

393-398) and just written in main text. I suggest that a summarize on

results in a table is more clear for readers.

Response : 

We agree that this would be clearer if presented in tables, and this has been done, by adding four more supplemental tables, and removing results from these paragraphs.

When writing these tables we have corrected some of the results (there are no significant effects of gender in the morning/evening comparisons for the healthy controls, and only four for the ADHD patients).

Concerning the effects of gender we have added a cautionary note with regard to the statistical results, noting that particularly for the healthy controls the groups are rather small, making the results more uncertain than for the groups containing both genders.

5.In graph theory, many measures are given, but the degree, path, acyclic

and spanning subgraph are not used in later analysis. If true, please

delete them.

Response :

We have altered the method section and removed the measures not used in the analyses. The definition of path is needed to define connected components and bridge. The definition of degree is needed to complete the definition of graph. We have however deleted the definition of tree and spanning graph. 

6.CT is cyclothymic temperament? Please identify the abbreviation while

using it first time.

Response :

Yes, CT is cyclothymic temperament. We have identified this abbreviation the first time it was used (line 312). 

7.Line 323, 39 should be 39 patients or something.

Response :

This has been amended to 39 patients.

8.The grammatical tense is not consistent, e.g., lines338 uses 'was', line

342 uses 'is', pleases unite them through the paper.

Response :

This has been altered, so the grammatical tense should now be consistent (past tense).

9.Line 344, what does the (1.34) mean.

Response :

This value is Cohens d, the sentence has been altered to make this clearer.

10.After tables, *p=0.05 should be p<0.05.

Response :

This has been altered.

11.Line 385, p=0.05 is not significant, pleases revise the description

Response :

This has been altered.

12.Lines 397-398, I didn't found the results corresponding to the

description.

Response :

The results are now presented in the supplemental tables 6 and 7.

13.Line 404, I know that 8/12 indicates only 8 measures are significant,

but it is really a strange description, can we just use 8?

Response :

We agree, and this has been altered.

14.Line 483, authors should present the current results and then compare

them to previous.

Response :

We have altered this part of the discussion and presented the results in more detail.

15.Line 500, is it a replicated description 'differently put; reduced

amplitude….. Reduced….'

Response :

Yes, it is just a different description of the same phenomenon, but we have altered the wording to : “or in other words”, to make this clearer..

16.In conclusion, I suggest authors don't use newly, because it is hard to

identify what is really new.

Response :

We have omitted this.

17.Finally, can author explain to me why we have a higher variability and

lower complexity in the evening, does it relate to our brain in some

aspects?

Response :

We have no clear explanation of this difference between morning and evening, but the inverse relation between variability and complexity is seen in different studies. See also response to reviewer 2.

Reviewer # 2 :

Major 1: During the construction of similarity graph, the similar was

detected by max(ux,uv)/min(ux,uv)<1.2. The authors explained this

definition with analogy to the tolerance in sample entropy. However, unlike

sample entropy where the tolerance range is a fix value (0.2*std), this

definition will lead to a flexible range depending on the amplitude of the

denominator. In other word, smaller signal value in the bottom will have

narrower tolerance range and harder to "be similar" to others. The authors

should explain why they use this criteria and what will be the consequence

in the following computation of graph measures.

Response : 

We have tried to explain this a bit more detailed in the method section of the paper. It is correct that with our definition of similarity the tolerance for similarity will depend on the value for the points in the time series in the neighbourhood of the index point, but we think it is more reasonable to use such a criterion compared to using a fixed value for the whole series (as in sample entropy). One problem with sample entropy is that it is sensitive to outliers, which will increase SD and thus increase the probability that two points are considered similar, giving a false impression of high regularity, and a general problem with motor activity studies with actigraphs is that SD is high (and sometimes very high). We therefore think it makes sense to let the tolerance range for similarity depend on the actual values of the points to be compared, rather than to be dependent on all the values of the series (that is on the SD). The consequence will be that number of points considered similar will be lower, thus avoiding ceiling effects, and probably increase the ability of the test to detect differences between groups. This will be the case for all the graph measures we use.

Minor 1: For readability, I suggest the authors to give a schematic

illustration for the construction of similarity graph on time-series. In

addition, authors can show example of actigraphs time-series. In addition,

different kinds of activity jumps detected by graph measures, degree,

bridge, and k-clique detect should be given to guide understanding.

Response :

We have added an illustration of a similarity graph (Fig 2) and an example of an actigraph time series (Fig 1). There are two kind of “jumps” detected by the graph analyses, components and bridges, these are illustated in Fig 2. In addition the figure also illustrates the concept of clique.

Minor 2: The discussion about diurnal fluctuations and weak circadian

rhythms, etc are too far away from the topic and the presented evidences,

given the fact that the authors cannot explain what is told by the bridge

changes.

Response :

We have substantially shortened and rewritten this part of the discussion.

Minor 3: Not clear why in healthy controls the evening registrations showed

higher variability and lower complexity compared to morning registrations.

authors can recheck.

Response :

We do not have a ready explanation of why there is such a difference between morning and evening registrations. Obviously we need more information on the diurnal regulation of motor activity and activation in general to understand we this is so. 

As we have noted in the response to reviewer 1, an inverse relation is seen between measures of variability and complexity is found also in other studies, for instance in a previous actigraph study from our group (Hauge ER, Berle JØ, Oedegaard KJ, Holsten F, Fasmer OB. Nonlinear analysis of motor activity shows differences between schizophrenia and depression: a study using Fourier analysis and sample entropy. PLoS One. 2011, 6 : e16291) and in a study on heart rate variability (Fasmer OB, Liao H, Huang Y, Berle JØ, Wu J, Oedegaard KJ, Wik G, Zhang Z. A naturalistic study of the effect of acupuncture on heart-rate variability. J Acupunct Meridian Stud. 2012, 5 : 15-20). The definition of these concepts (SD and sample entropy) are clearly different, but as this study also shows they are in some way related.

---

## [Decision Letter · Decision Letter 1]

26 Oct 2020

Diurnal variation of motor activity in adult ADHD patients analyzed with methods from graph theory

PONE-D-20-19305R1

Dear Dr. Fasmer,

We’re pleased to inform you that your manuscript has been judged scientifically suitable for publication and will be formally accepted for publication once it meets all outstanding technical requirements.

Kind regards,

Pan Lin

Academic Editor

PLOS ONE

Additional Editor Comments (optional):

Reviewers' comments:

Reviewer's Responses to Questions

**Comments to the Author**

1. If the authors have adequately addressed your comments raised in a previous round of review and you feel that this manuscript is now acceptable for publication, you may indicate that here to bypass the “Comments to the Author” section, enter your conflict of interest statement in the “Confidential to Editor” section, and submit your "Accept" recommendation.

Reviewer #1: All comments have been addressed

Reviewer #2: All comments have been addressed

2. Is the manuscript technically sound, and do the data support the conclusions?

Reviewer #1: Yes

Reviewer #2: Yes

3. Has the statistical analysis been performed appropriately and rigorously? 

Reviewer #1: Yes

Reviewer #2: Yes

4. Have the authors made all data underlying the findings in their manuscript fully available?

Reviewer #1: Yes

Reviewer #2: No

5. Is the manuscript presented in an intelligible fashion and written in standard English?

Reviewer #1: Yes

Reviewer #2: Yes

6. Review Comments to the Author

Reviewer #1: I recommend the publication because the authors have addressed all my comments. But I still have two minor requests:

1. There is no any label in figures. 1 and 2. In figure 1, three actigraph recordings are for ADHD, normal and control ADHD? please clearly state it. In figure 2, it is more clear if showing the value of max (xu , xv) / min (xu , xv).

2. In method, please first show the construction of Similarity graph and then show the Graph theory.

Reviewer #2: (No Response)

7. PLOS authors have the option to publish the peer review history of their article (what does this mean?). If published, this will include your full peer review and any attached files.

Reviewer #1: No

Reviewer #2: No

---

## [Editor Report · Acceptance letter]

29 Oct 2020

PONE-D-20-19305R1 

Diurnal variation of motor activity in adult ADHD patients analyzed with methods from graph theory 

Dear Dr. Fasmer:

I'm pleased to inform you that your manuscript has been deemed suitable for publication in PLOS ONE. Congratulations! Your manuscript is now with our production department. 

Kind regards, 

on behalf of

Dr. Pan Lin 

Academic Editor

PLOS ONE